# Application of SPRi Biosensors for Determination of 20S Proteasome and UCH-L1 Levels in the Serum and Urine of Transitional Bladder Cancer Patients

**Anna Sankiewicz [1], Tomasz Guszcz [2] and Ewa Gorodkiewicz [1,\*]**

[1] Bioanalysis Laboratory, Faculty of Chemistry, University of Bialystok, Ciolkowskiego 1K, 15-245 Bialystok, Poland; ania@uwb.edu.pl

[2] Department of Urology, J. Sniadecki Provincial Hospital of Bialystok, M. Sklodowskiej-Curie 26, 15-950 Bialystok, Poland; tomasz.guszcz@o2.pl

\* Correspondence: ewka@uwb.edu.pl

**Abstract:** The ubiquitin–proteasome system (UPS) participates in the degradation of proteins which play an important role in regulating the cell cycle, apoptosis, and angiogenesis, as well as in the immune system. These processes are important in carcinogenesis. Transitional cell carcinoma (TCC) is one of the predominant types of bladder cancer. The relationship between the ubiquitin–proteasome system and cancer progression has become a topic of increasing interest among researchers. In this work, we propose an application of surface plasmon resonance imaging (SPRi)-based biosensors for the detection of 20S proteasome and ubiquitin carboxy-terminal hydrolase L1 (UCH-L1) in the blood serum and urine of patients with TCC. The aim of the study was to determine 20S proteasome and UCH-L1 concentrations and to correlate the results with clinicopathological parameters. The group of subjects consisted of 82 patients with confirmed TCC, in addition to a control group of 27 healthy volunteers. It was found that 20S proteasome and UCH-L1 concentrations were significantly elevated in both the serum and urine of TCC patients, compared with the healthy subjects. There was a correlation between 20S proteasome concentrations in serum and urine, as well as between serum proteasome and UCH-L1 concentration. The SPRi biosensor sensitive to 20S proteasome using PSI inhibitor as the receptor, and the SPRi biosensor sensitive to the UCH-L1 protein using the protein-specific antibody as the receptor is suitable for the determination of 20S proteasome and UCH-L1 in body fluids and can serve as useful tools in the investigation of cancer biomarkers.

**Keywords:** proteasome; ubiquitin carboxy-terminal hydrolase L1 (UCH-L1); bladder cancer; surface plasmon resonance imaging biosensor

## 1. Introduction

Urinary bladder cancer (BCa) is the 10th most common cancer in the world, with an estimated 549,393 diagnosed cases and 200,000 deaths annually. It is the 6th most common malignancy in males and the 17th in females, with age-standardised risks of 9.6% and 2.4%, respectively [1].

This disease can present as non-muscle-invasive bladder cancer (NMIBC), muscle-invasive bladder cancer (MIBC), or the metastatic form of the disease. Tobacco smoking is the most important risk factor for BCa: it is responsible for approximately half of BCa cases and is associated with poor oncological outcomes for both NMIBC and MIBC [2]. Stopping smoking reduces the risk of developing BC by almost 40% within 5 years [3]. Bladder cancer remains a highly prevalent and lethal malignancy. The optimal selection of treatment depends on early diagnosis and accurate staging and grading.

Biomarkers are critical in routine clinical practice. They serve as indicators for the detection of bladder carcinoma and for the prediction of its recurrence and progression. A number of bladder cancer markers are described in the literature [4].

Numerous reports have demonstrated that an aberrant process of the ubiquitin–proteasome system results in the disturbance of protein degradation. This disturbance can lead to tumourigenesis [5,6]. Increased concentrations of circulating proteasomes have been demonstrated in patients with multiple myeloma [7], acute leukaemia [8], malignant melanoma [9], and various solid tumours [10,11]. Studies have demonstrated that ubiquitin carboxy-terminal hydrolase L1 (UCH-L1) is implicated in oncogenesis, tumour cell invasion, and metastasis [12–14]. The expression of UCH-L1 has been reported in a variety of cancers [15]. The literature data suggest that UCH-L1 may play a role as a prognostic marker for tumour growth and progression [16–19].

The relationship between the ubiquitin–proteasome system and cancer progression has become a topic of increasing interest. The ubiquitin–proteasome system (UPS) is involved in intracellular protein degradation and the regulation of many cellular processes such as the cell cycle, induction of immune response, and gene expression [20–22]. These processes are highly relevant to tumour progression and carcinogenesis [23].

The UPS has several components, including ubiquitin, 26S proteasome, ubiquitin-activating enzyme (E1), ubiquitin-conjugating enzyme (E2), ubiquitin-ligating enzyme (E3), and deubiquitinating enzymes (DUBs) [6,20].

The central element of this system is the 26S proteasome. This is a multi-subunit proteolytic complex, consisting of the 19S regulatory particle and the 20S core particle. The 19S subunit is responsible for substrate recognition, deubiquitination, and unfolding, while the 20S proteasome is responsible for proteolytic activity and protein degradation. The 20S proteasome (M ~700 kDa) has a cylindrical structure and is composed of two outer rings and two inner rings. The outer rings contain $\alpha$-type subunits, whose function is to control the entry of the substrate proteins into the central catalytic chamber and bind the regulators. The inner rings contain $\beta$-type subunits. Three of the $\beta$-subunits, $\beta1$, $\beta2$, and $\beta5$, are catalytically active and are responsible, respectively, for the caspase-like (or peptidylglutamyl-peptide), trypsin-like, and chymotrypsin-like hydrolysing functions [24].

The processes of ubiquitination and deubiquitination play a very important role in protein degradation in the proteasome. Prior to the entry of protein into the proteasome, the deubiquitinating enzymes (DUBs) remove ubiquitin from the substrate proteins. Ubiquitin carboxy-terminal hydrolase L1 (UCH-L1), also known as PGP9.5, is one of the deubiquitinating enzymes. UCH-L1 belongs to the family of peptidases. This enzyme hydrolyses a peptide bond at the C-terminal glycine of ubiquitin [25] and is a mono-ubiquitin stabiliser [26].

As the levels of 20S proteasome and UCH-L1 can be detected in serum [7,10,27,28], they can be used as potential cancer biomarkers. ELISA is the preferred method for the determination of 20S proteasome and UCH-L1 concentrations. Although it is a reliable technique with good sensitivity and selectivity, it is relatively time- and labour consuming. SPRi may be an alternative technique to traditional immuno-based assays such as ELISA. Both methods have advantages and disadvantages. SPR has been found to be comparable to ELISA in terms of sensitivity and specificity. In SPR, 20S proteasome and UCH-L1 concentrations can be detected in real time using only a single antibody or inhibitor [27,29–31]. Unlike ELISA, SPRi does not require a label or an additional reagent. An SPR biosensor requires very low amounts of reagents and samples and does not require any special treatment [32–35]. It enables the rapid and precise determination of 20S proteasome and UCH-L1 concentrations.

The surface plasmon resonance (SPR) method is an optical technique that measures the changes of refractive index caused by molecules bound to a metal surface. Conventional SPR sensors measure the reflectance as a function of the angle of incident light.

The SPR imaging version (SPRi) eliminates the complexity of scanning the angle. The measurements are made at a particular angle of incident light. The reflected light is collected using a CCD and presented as an image. The angle at which the measurement is carried out lies in the region of linear decrease in reflectance. The changes in light intensity are proportional to the mass of biomolecules attached to the biosensor surface.

A very important part of an SPRi system is a sensor chip with a sensitive recognition element. This is usually a glass chip coated with an inert metal (e.g., gold with thickness 50 nm).

The immobilisation of biocomponents on a chemically modified gold layer occurs via covalent bonds, hydrophobic interaction, or adsorption. This is an important step in the manufacture of a biosensor because it has an influence on the biosensor's efficiency.

The aim of this study was to use SPRi biosensors to determine concentrations of 20S proteasome and UCH-L1 in the blood serum and urine of patients with bladder cancer and to correlate the results with clinical–pathological parameters. To our knowledge, this is the first paper presenting data on 20S proteasome and UCH-L1 protein obtained from measurements in the serum and urine of patients with bladder cancer. The study assessed the possible effectiveness of these markers for the diagnosis of invasive bladder cancer and its superficial form with an increased risk of progression. Quantification of the components of the ubiquitin–proteasome system can lead to a more accurate prognosis in bladder cancer and can be useful for identifying high-risk patients and determining the optimal duration of treatment.

## 2. Materials and Methods

### 2.1. Reagents

For this study, 20S proteasome (mammalian) (AFFINITI Research Products Ltd., Mamhead, UK), PSI (Z-Ile-Glu(OBut)-Ala-Leu-H), (BIOMOL, Lörrach, Germany), recombinant human UCH-L1 protein, rabbit monoclonal mouse IgG2A antibody specific for human UCH-L1(R&D Systems, Minneapolis, MN, USA), cysteamine hydrochloride, N-ethyl-N'-(3-dimethylaminopropyl) carbodiimide (EDC) (Sigma, Steinheim, Germany) and N-hydroxysuccinimide (NHS) (Aldrich, Munich, Germany) were used. HBS-ES solution pH = 7.4 (0.01 M HEPES, 0.15 M sodium chloride, 0.005% Tween 20, 3 mM EDTA), photopolymer ELPEMER SD 2054, hydrophobic protective paint SD 2368 UV SG-DG (Peters, Kempen, Germany), phosphate buffered saline (PBS) pH = 7.4, and carbonate buffer pH = 8.5 (BIOMED, Lublin, Poland) were used as received. Aqueous solutions were prepared with MilliQ water (Simplicity®MILLIPORE).

### 2.2. Patients

The samples were obtained from patients with TCC who were observed at the J. Sniadecki Provincial Hospital of Bialystok (Bialystok, Poland). The subjects were divided into malignant and control groups. Urine and serum samples were obtained from patients already diagnosed (by cystoscopy or computer tomography) with bladder cancer. Individuals with additional malignant or inflammatory disease (also negative urine cultures) were not included in the study. The cancer diagnosis was determined by histological examination of tumour specimens obtained from transurethral resection or cystectomy. Finally, the malignant group consisted of 82 patients with confirmed TCC. Clinical parameters, including stage, grade, size, the tendency to recur, the pattern of growth, and multifocal nature, were determined. The stage and grade were based on the TNM classification, which was approved by the Union for International Cancer Control (UICC) in 2009 and updated in 2017 (eighth edition) but with no changes in relation to bladder tumours. Patients with recurrent tumours receiving intravesical chemotherapy or BCG therapy were not included in the study. Clinical characteristics of the patients are given in Table 1. The control group included 27 healthy volunteers (less than 68 years of age) from the Blood Donor Centre in Bialystok, Poland.

**Table 1.** Demographic and clinicopathological characteristics of patients.

| Variable | Range | Number of Patients |
|---|---|---|
| Age (year) | <65 | 34 |
| | >65 | 48 |
| Gender | Women | 23 |
| | Men | 59 |
| Tumour stage | Superficial (Ta + T1) | 51 |
| | Invasive (T2 + T3 + T4) | 31 |
| Tumour grade | Low grade | 35 |
| | High grade | 47 |
| Tumour size (mm) | <30 | 49 |
| | >30 | 33 |
| Recurrence | Primary | 35 |
| | Recurrent | 47 |
| Multiplicity | Single | 46 |
| | Multiply | 36 |

Approval for this study was obtained from the Bioethics Committee of the Medical University of Bialystok (R-I-002/409/2014, Bialystok, Poland), and written informed consent was obtained from all patients and donors.

*2.3. Preparation of Biological Samples*

Blood samples were obtained from the median cubical vein. Serum was prepared according to standard protocols. Urine samples were centrifuged at $1850\times g$ for 15 min, and the supernatant was separated. Finally, the sample was filtered through a paper filter of medium density. The urine and serum samples were frozen immediately and maintained at $-70\,^{\circ}$C. For the determination of concentrations of 20S proteasome and UCH-L1, the prepared serum samples were diluted tenfold with phosphate-buffered saline (PBS). Urine samples were not diluted.

*2.4. Procedure for Determination of Concentrations with SPRi Biosensors*

2.4.1. Biosensor Preparation

Gold chips were manufactured as described in a previous paper [34]. The gold surface of the chip was covered with photopolymer and hydrophobic paint. The chip has 9 sites each with 12 free gold surfaces. Using this chip, nine different solutions can be simultaneously measured without mixing the tested solutions. Overall, 12 single SPRi measurements can be performed from one sample [34].

The chips were then immersed in 20 mM of cysteamine ethanolic solution for at least 2 h. They were then rinsed with ethanol and water and dried under a stream of nitrogen. The next step was immobilisation of the receptor. The receptors used were PSI inhibitor for 20S proteasome, and specific rabbit monoclonal antibody for the protein UCH-L1. PSI inhibitor at a concentration of 80 nM and antibody solution in a PBS buffer (10 μg/mL) were activated with NHS (50 mM) and EDC (200 mM) in a carbonate buffer (pH = 8.5) environment. Then, the activated receptors were placed on a thiol (cysteamine) modified surface and incubated at 37 °C for 1 h. The detailed procedures for biosensor construction and the preparation of calibration curves are described in previous papers [29,30]. The biosensors used are shown in Figure 1.

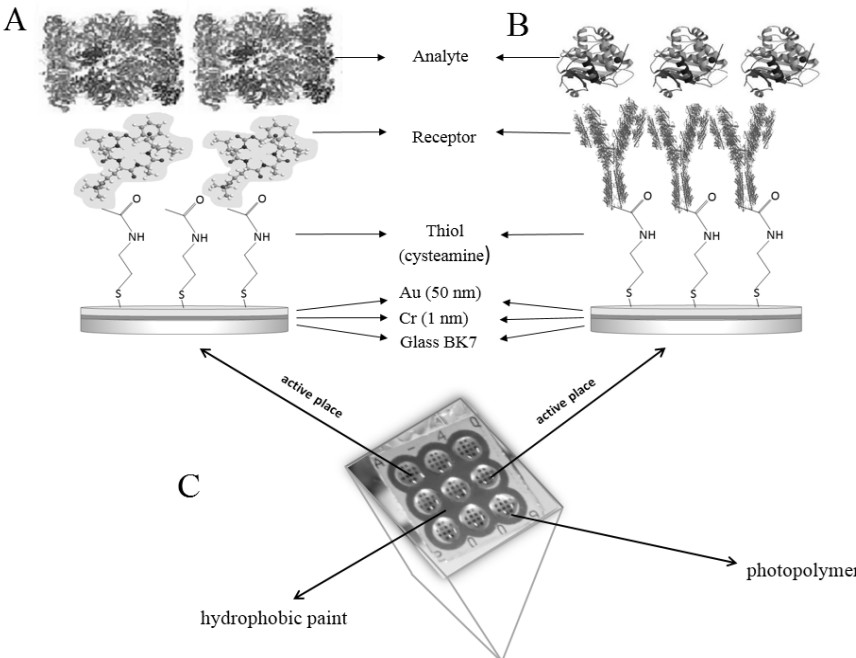

**Figure 1.** Illustration of the biosensors. Scheme of the active part of the biosensor for 20S proteasome (**A**) and UCH-L1 (**B**). Picture of the chip on the prism (**C**).

### 2.4.2. SPRi Measurements

SPRi measurements were performed using a specially made apparatus and biosensors which have been successfully used and described previously [35]. The samples of serum and urine were placed directly on the prepared biosensor for 10 min to allow interaction with the receptor. The volume of the sample applied to each measuring field was 3 μL. After this time, the biosensor was washed with water and HBS-ES buffer solution at pH = 7.4 (0.01 M 4-(2-hydroxyethyl) piperazine-1-ethanesulfonic acid, 0.15 M sodium chloride, 0.005% Tween 20, 3 mM EDTA) to remove unbound molecules from the surface.

The SPRi signal was measured at a constant angle of incident light. The image was recorded twice: after immobilisation of the receptor and then after interaction with the sample containing the analyte. The SPRi signal, which is proportional to the quantity of coupled biomolecules, was obtained for each spot separately, as there was a difference between the signals before and after interaction with the biomolecule.

Non-specific binding was eliminated by a background correction. This is the signal difference between the site of the receptor–analyte complex and a receptor-free site treated with a natural sample.

### 2.5. Statistical Analysis

The results are presented as mean or median $\pm$ standard deviation. Statistical analyses were performed using the unpaired *t*-test (for normally distributed variables) or the Mann–Witney U test (for variables that failed the normality test), and the *p*-value was automatically calculated ($p < 0.05$ was considered to indicate a statistically significant difference). The Kolmogorov–Smirnov test was performed to test the assumption of normal distribution. Correlations were examined by linear regression using the Spearman or Pearson test (for normal distribution). The receiver operating characteristic curves with optimal cut-off points were calculated. For the cut-off points, sensitivity, specificity, and positive and negative predictive values were calculated. The statistical analysis was carried out with the use of PQ Stat v.1.6.4.

### 3. Results

The 20S proteasome and UCH-L1 concentrations in the blood serum and urine of patients with bladder cancer and healthy donors were measured using the SPRi biosensor, and the results are presented in Table 2 and Figure 2.

**Table 2.** Concentrations of 20S proteasome and UCH-L1 in the examined samples.

| | Bladder Cancer Patients | | Healthy Donors | | | |
|---|---|---|---|---|---|---|
| | Average | Median | Average | Median | *p* | Test |
| Proteasome serum [µg/mL] | 15.45 | 15.13 | 2.72 | 2.89 | <0.0001 | T-Student for independent |
| Proteasome urine [µg/mL] | 1.69 | 1.81 | 0.26 | 0.24 | <0.0001 | U-Mann Whitney |
| UCHL-1 serum [ng/mL] | 4.97 | 4.72 | 0.45 | 0.45 | <0.0001 | U-Mann Whitney |
| UCHL-1 urine [ng/mL] | 0.63 | 0.66 | 0.18 | 0.18 | <0.0018 | U-Mann–Whitney |

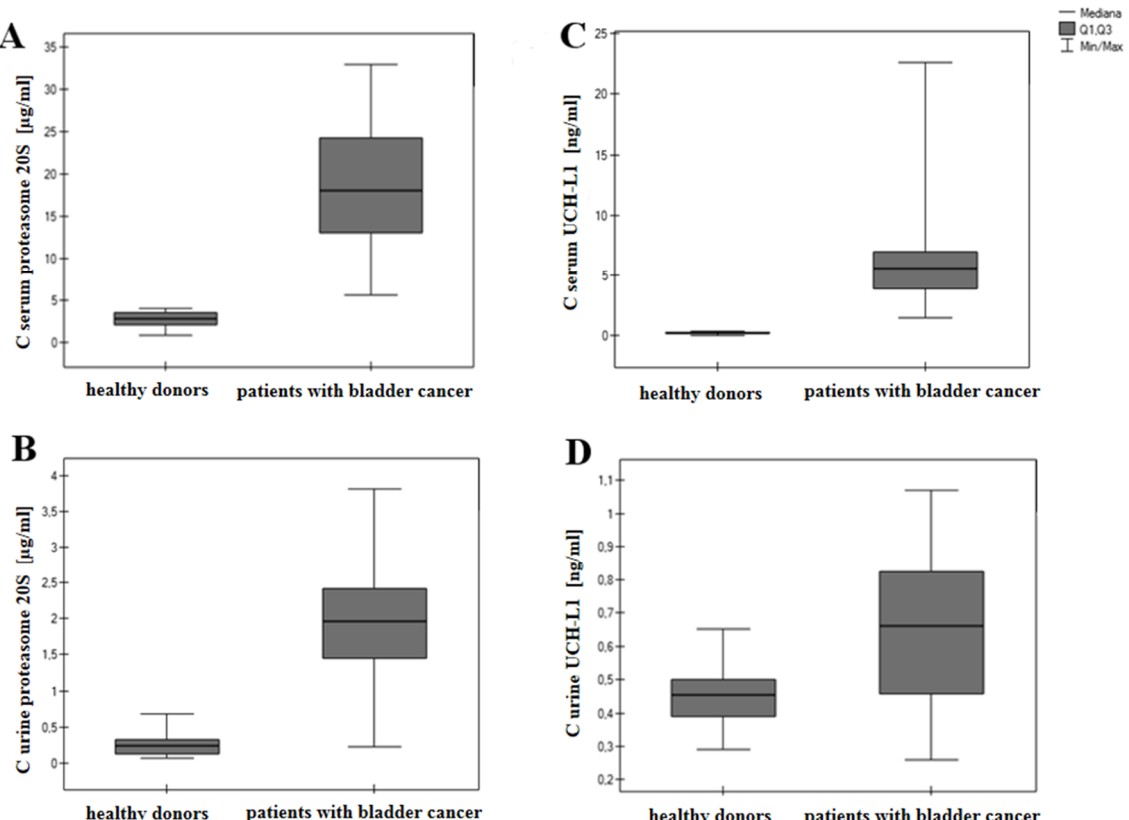

**Figure 2.** Concentrations of 20S proteasome and UCH-L1 in the serum and urine of bladder cancer patients and healthy subjects. (**A**) and (**C**) in serum samples and (**B**) and (**D**) in urine samples.

A significant difference in serum and urine concentrations of 20S proteasome and UCH-L1 was observed between patients with bladder cancer and healthy subjects. The 20S proteasome concentrations in the serum of bladder cancer patients and healthy donors were in the ranges 3.75–36.42 µg/mL and 0.89–4.01 µg/mL, respectively, while the urine 20S proteasome concentration was in the range 0.32–3.81 µg/mL for cancer patients and 0.07–0.68 µg/mL for healthy donors. The serum UCH-L1 concentration was in the range 1.47–22.61 ng/mL for

cancer patients and 0.07–0.35 ng/mL for healthy donors. The urine UCH-L1 concentration was in the range 0.32–3.81 μg/mL for cancer patients and 0.29–0.65 ng/mL for healthy donors.

The 20S proteasome concentrations in serum and urine were significantly higher in patients with bladder cancer (median values 15.13 μg/mL and 1.81 μg/mL, $p < 0.001$) than in the controls (median 2.89 μg/mL and 0.24 μg/mL, $p < 0.001$). Similarly, the UCH-L1 concentration in serum and urine was significantly elevated in subjects with bladder cancer (median 4.72 ng/mL and 0.66 ng/mL, $p < 0.001$) versus healthy subjects (median 0.45 ng/mL and 0.18 ng/mL, $p < 0.001$). The results are presented in Figure 2.

The 20S proteasome and UCH-L1 concentration results were analysed in terms of various cancer parameters. The following factors were taken into account: the recurrent nature of the tumour, tumour stage, tumour grade, size, and multiplicity of the tumour. The serum and urine concentrations of 20S proteasome and UCH-L1 in correlation with clinicopathological characteristics are shown in Tables 3 and 4.

**Table 3.** Diagnostic characteristics of serum 20S proteasome and UCH-L1 concentration ratio compared with various clinicopathological parameters.

| Parameter | Proteasome 20S Concentration [μg/mL] | | | UCH-L1 Concentration [ng/mL] | | |
|---|---|---|---|---|---|---|
| | Range | Median | *p*-Value | Range | Median | *p*-Value |
| Primary/recurrent | | | | | | |
| Primary (35) | 7.78–36.42 | 19.81 | 0.0258 | 1.47–22.61 | 6.3 | 0.0621 (NS) |
| Recurrent (47) | 3.75–32.91 | 15.92 | | 1.54–11.51 | 5.5 | |
| Multiplicity | | | | | | |
| Single (46) | 3.75–36.42 | 18.08 | 0.6458 (NS) | 1.54–11.32 | 5.36 | 0.0329 |
| Multiply (36) | 5.64–32.91 | 18.94 | | 1.47–22.61 | 6.7 | |
| Stage | | | | | | |
| Non-muscle invasive (51) (Ta + T1) | 3.75–32.91 | 16.27 | 0.0264 | 1.47–11.51 | 4.99 | 0.0001 |
| Muscle invasive (31) (T2 + T3 = T4) | 5.64–36.42 | 23.54 | | 2.74–22.61 | 6.95 | |
| Grade | | | | | | |
| Low-grade (34) | 3.75–28.99 | 15.06 | 0.0036 | 1.47–9.85 | 4.65 | 0.0001 |
| High-grade (48) | 5.64–36.42 | 20.07 | | 2.74–22.61 | 6.62 | |
| Size (mm) | | | | | | |
| <30 (49) | 6.19–30.56 | 16.62 | 0.0668 (NS) | 1.54–22.61 | 5.6 | 0.1761 (NS) |
| >30 (33) | 3.75–36.42 | 19.68 | | 1.47–12.34 | 6.41 | |

NS—no statistically significant values.

Concentrations of 20S proteasome and UCH-L1 were detectable in all serum and urine samples. There was no significant difference in serum and urine concentrations of 20S proteasome and UCH-L1 between females and males, nor was there any correlation of these concentrations with age, either in the control group or in patients with bladder cancer. The median values of serum 20S proteasome and UCH-L1 concentration were statistically higher in patients with muscle-invasive tumour than in those with non-muscle-invasive tumour ($p = 0.0264$ for 20S proteasome, $p = 0.0001$ for UCH-L1). The concentrations were also statistically higher in patients with high-grade tumour ($p = 0.0036$ for 20S proteasome, $p = 0.0001$ for UCH-L1) than in those with low-grade tumour. Similar dependencies were found for 20S proteasome concentration in urine (higher for muscle-invasive tumour and high-grade tumour; $p = 0.0448$ and $p = 0.0033$, respectively). Concentrations of 20S proteasome in the serum and urine of patients with a primary tumour were higher than for patients with a recurrent tumour ($p = 0.0258$ for serum, $p = 0.0384$ for urine). Values of

UCH-L1 concentration were significantly elevated in the serum of patients with multiple tumours, compared with patients with a single tumour ($p = 0.0329$), and in the urine of patients with tumour size greater than 30 mm ($p = 0.0184$). There were no statistically significant correlations between serum and urinary UCH-L1 concentrations and other clinicopathological parameters.

**Table 4.** Diagnostic characteristics of urine 20S proteasome and UCH-L1 concentration ratio compared with various clinicopathological parameters.

| Parameter | Proteasome 20S Concentration [µg/mL] | | | UCH-L1 Concentration [ng/mL] | | |
|---|---|---|---|---|---|---|
| | **Range** | **Median** | *p*-**Value** | **Range** | **Median** | *p*-**Value** |
| | Primary/recurrent | | | | | |
| Primary (35) | 0.32–3.62 | 2.25 | 0.0446 | 0.28–1.07 | 0.71 | 0.0892 |
| Recurrent (47) | 0.38–3.81 | 1.81 | | 0.26–1.05 | 0.67 | (NS) |
| | Multiplicity | | | | | |
| Single (46) | 0.38–3.81 | 2.08 | 0.5656 | 0.28–1.07 | 0.7 | 0.1722 |
| Multiply (36) | 1.13–3.51 | 1.85 | (NS) | 0.26–1.05 | 0.67 | (NS) |
| | Stage | | | | | |
| Non-muscle invasive (51) (Ta + T1) | 0.38–3.81 | 1.82 | 0.0448 | 0.27–1.05 | 0.66 | 0.0574 (NS) |
| Muscle invasive (31) (T2 + T3 = T4) | 1.23–3.62 | 2.25 | | 0.26–1.07 | 0.77 | |
| | Grade | | | | | |
| Low-grade (34) | 0.38–3.38 | 1.7 | 0.0033 | 0.28–1.07 | 0.67 | 0.4409 |
| High-grade (48) | 1.13–3.81 | 2.21 | | 0.26–1.05 | 0.73 | (NS) |
| | Size (mm) | | | | | |
| <30 (49) | 0.38–3.81 | 1.81 | 0.2258 | 0.26–1.05 | 0.66 | 0.0184 |
| >30 (33) | 0.42–3.62 | 2.21 | (NS) | 0.28–1.07 | 0.85 | |

NS—no statistically significant values.

### 3.1. ROC Analysis

ROC curve analyses demonstrated that serum 20S proteasome and UCH-L1 levels were capable of distinguishing patients with muscle-invasive tumour from patients with non-muscle-invasive tumour, with areas under the curve of 0.64 ($p = 0.029$) and 0.74 ($p = 0.003$), respectively (Figure 3A,B). Furthermore, serum 20S proteasome and UCH-L1 levels were capable of distinguishing patients with high-grade tumour from patients with low-grade tumour, with areas under the curve of 0.65 ($p = 0.019$) and 0.64 ($p = 0.020$), respectively (Figure 3C,D).

Additionally, serum levels of proteasome and UCH-L1 in patients with primary and recurrent tumour were compared. The ROC curve for discriminating between those groups yielded an AUC of 0.62 ($p = 0.49$) for serum 20S proteasome and 0.53 ($p = 0.62$) for serum UCH-L1 (Figure 3E,F). For each variable cut-off level, positive predictive value (PPV) and negative predictive value (NPV) were calculated. The results are presented in Table 5.

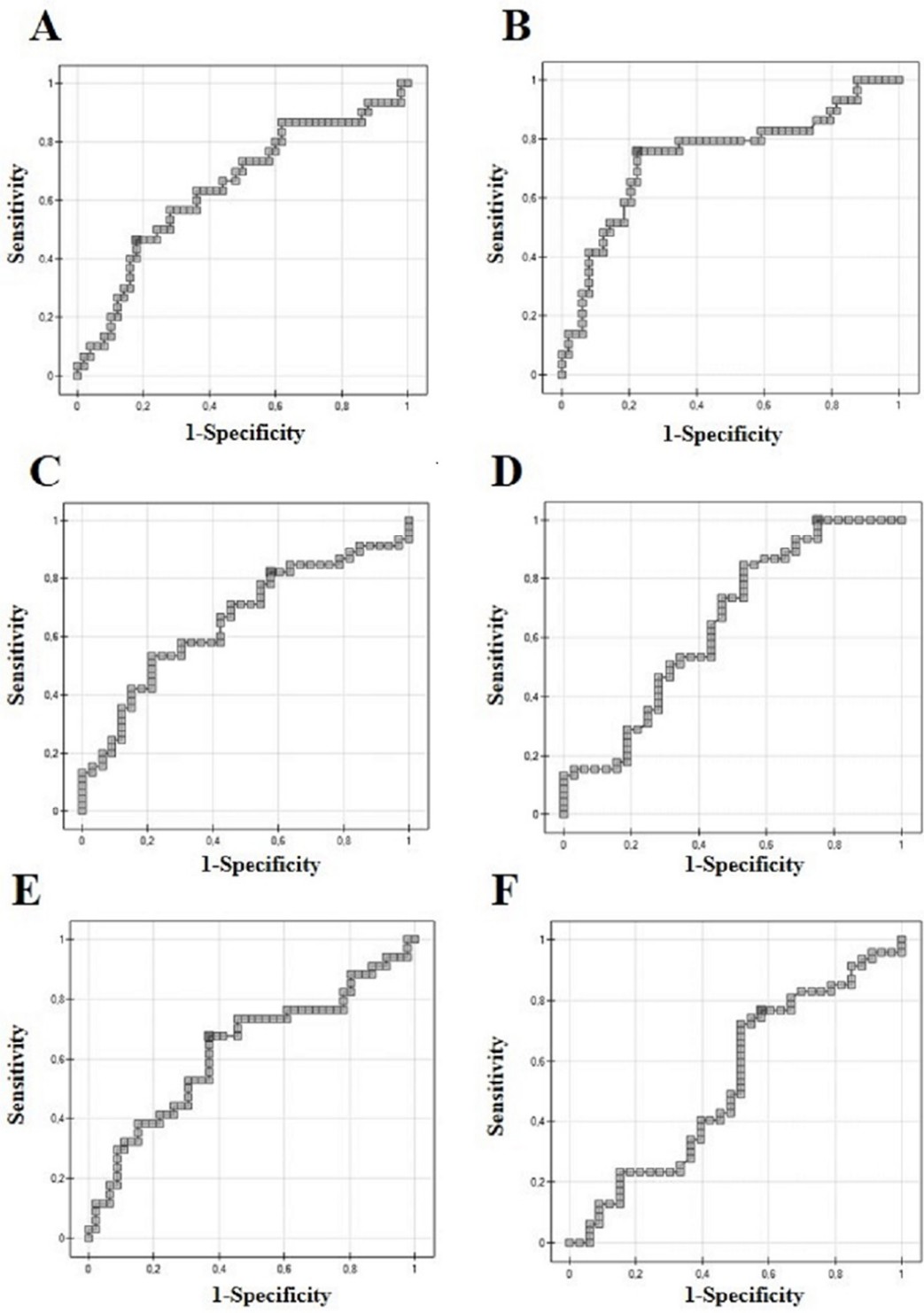

**Figure 3.** ROC curves for 20S proteasome in serum as diagnostic tests for (**A**) muscle-invasive tumour, (**B**) high-grade tumour, (**C**) recurrent tumour, and for UCH-L1 in serum as diagnostic tests for (**D**) muscle-invasive tumour, (**E**) high-grade tumour, (**F**) recurrent tumour.

**Table 5.** Diagnostic efficiency of serum 20S proteasome and UCH-L1.

| | AUC | *p* | Sensitivity | Specificity | PPV | NPV | Cut-Off |
|---|---|---|---|---|---|---|---|
| Proteasome 20S Muscle-invasive (Figure 2A) | 0.64 | 0.029 | 46 | 82 | 61 | 71 | 21.79 |
| Proteasome 20S High-grade (Figure 2B) | 0.65 | 0.019 | 82 | 42 | 66 | 63 | 13.54 |
| Proteasome 20S Recurrence (Figure 2C) | 0.62 | 0.49 | 67 | 63 | 57 | 72 | 17.48 |
| UCH-L1 Muscle-invasive (Figure 2D) | 0.74 | 0.0003 | 75 | 77 | 66 | 84 | 6.27 |
| UCH-L1 High-grade (Figure 2E) | 0.64 | 0.02 | 100 | 25 | 65 | 100 | 2.74 |
| UCH-L1 Recurrence (Figure 2F) | 0.53 | 0.62 | 76 | 42 | 65 | 56 | 4.51 |

AUC—area under the curve, PPV—positive predictive value, NPV—negative predictive value.

### 3.2. Correlations of 20S Proteasome and UCH-L1

Correlation analysis between 20S proteasome and UCH-L1 concentrations was performed using the Pearson test (for normally distributed data) or the Spearman test (for non-normally distributed data). The levels of urine 20S proteasome correlated positively with the levels of 20S proteasome in serum (r = 0.64; Pearson test) (Figure 4A). No significant correlation was found between urine UCH-L1 levels and serum UCH-L1 levels (r = 0.08; Spearman test).

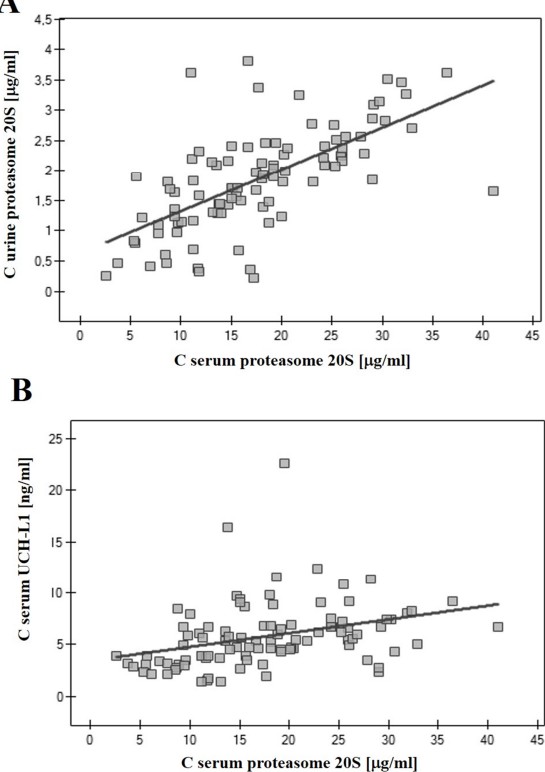

**Figure 4.** Pearson's correlation analysis between urine and serum 20S proteasome concentration (r = 0.64) (**A**), and Spearman's correlation analysis between serum UCH-L1 and serum 20S proteasome concentration (r = 0.47) (**B**).

Spearman tests were also performed for linear correlation between UCH-L1 and 20S proteasome concentrations in serum and between UCH-L1 and 20S proteasome concentrations in urine. A positive correlation (r = 0.47) was observed between UCH-L1 and 20S proteasome in serum only (Figure 4B).

### 3.3. UCH-L1 to 20S Proteasome Serum Concentration Ratio

The UCH-L1 to 20S proteasome serum concentration ratio was calculated for each case, and an ROC curve was obtained to distinguish patients with bladder cancer and healthy individuals (Figure 5).

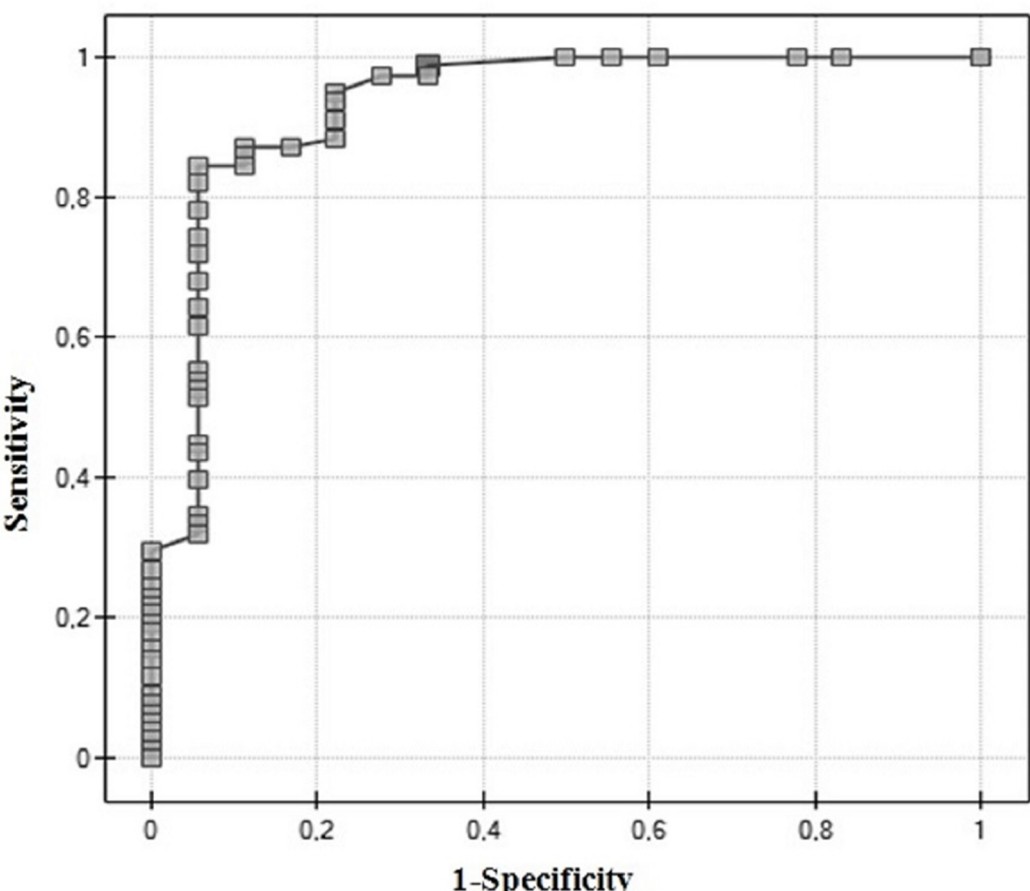

**Figure 5.** ROC curves for concentration ratio of UCH-L1 to 20S proteasome in serum as diagnostic tests for bladder cancer.

The ROC curve yielded an AUC of 0.93 ($p < 0.0001$), with sensitivity 98%, specificity 66%, PPV 92%, and NPV 92%. The calculated cut-off point was 0.09.

### 4. Discussion

Research into new malignancy markers is crucial for the development of clinical oncology. In this study, the focus is on the 20S proteasome, which is part of the ubiquitin–proteasome system, and the protein UCH-L1, which belongs to the family of deubiquitinating enzymes (DUBs). The measurements presented in the paper are the first available data on concentrations of 20S proteasome and UCH-L1 protein, and their mutual correlation in the serum and urine of bladder cancer patients.

The ubiquitin–proteasome system (UPS) regulates almost all cellular activities, including cell cycle progression, DNA replication and repair, transcription, protein quality control, immune response, and apoptosis [36]. UCH-L1 plays an important role in proteasome activity by cleaving the ubiquitin C-terminal from the ubiquitinated protein to facilitate protein degradation by the 20S proteasome. The serum and urine concentrations of both

substances were determined and correlated with clinicopathological parameters of patients with bladder tumour (transitional cell carcinoma, TCC).

It was observed that proteasome and UCH-L1 concentrations were significantly elevated in both the serum and urine of TCC patients. Levels of 20S proteasome were six times higher in serum and seven times higher in urine, in TCC patients, as compared with healthy donors. UCH-L1 levels were eleven times higher in serum and three times higher in urine in TCC patients than in healthy donors (Table 2). In addition, there was a correlation between proteasome concentrations in serum and urine, as well as between serum proteasome concentration and UCH-L1 concentration (Figure 4).

The results obtained are consistent with similar previous studies. It has been reported that serum 20S proteasome concentration is elevated in the case of, for example, renal cell carcinoma [37], multiple myeloma [38], breast cancer [39], ovarian cancer [11], and malignant melanoma [40]. Benign conditions, including autoimmune, vascular, and pulmonary conditions, can also alter circulating proteasome levels [41,42]. Gruba et al. reported increased proteasome activity in TCC [43]. It is hypothesised that the 20S proteasome produces activity-related or tumour-associated effects [39].

There have been many reports confirming the relationship between UCH-L1 and cancer. It has been found to be overexpressed in prostate [44], renal [17], colorectal [19], and pancreatic [18] cancers, uterine serous cancer [45], and lymphoblastic leukaemia [46]. It has been confirmed that cancer cells are oxidative stressed. They produce more reactive oxygen species (ROS) and antioxidant systems are suppressed [47]. Oxidative stress can increase UCH-L1, proteasome subunit alpha, and proteasome activity [48,49]. The UPS (ubiquitin–proteasome system) removes small damaged proteins and protein aggregates and protects tissues from oxidative damage [50]. The UPS is responsible for the selective degradation of proteins engaged in cellular metabolism, for example, in regulating the cell cycle, apoptosis, and gene expression [21,51]. Higher expression of proteasome and its activity in malignant cells increase the degradation of cycle-controlling proteins and apoptosis, which leads to neoplasm growth. Increased proteasome levels in plasma result from reinforced secretion and disintegration of neoplasm cells.

With respect to clinicopathological features, serum 20S proteasome concentration was significantly higher for high-grade than for low-grade TCC ($p = 0.0036$). It was also higher for primary than for recurrent TCC, and for muscle-invasive than for non-muscle-invasive tumours. Serum UCH-L1 concentration was significantly higher for high-grade than for low-grade TCC ($p = 0.0001$) and for muscle-invasive than for non-muscle-invasive TCC ($p = 0.0001$), and was higher for multiple than for single TCC. The strong diagnostic potential of serum UCH-L1 concentration for distinguishing between muscle-invasive and non-muscle-invasive TCC is demonstrated by the corresponding ROC curve (Figure 3D and Table 5), with an AUC of 0.74. The cut-off value is 6.27 ng/mL. These results correspond with those reported by Stoebner et al., in which variables representing tumour size, differentiation, and proliferation were associated with elevated 20S proteasome levels [40].

The results for urine 20S proteasome concentration (Table 4) were similar to those for serum: the concentration was significantly higher for high-grade than for low-grade TCC ($p = 0.0033$) and was higher for primary than for recurrent TCC and for muscle-invasive than for non-muscle-invasive tumours. Additionally, urine UCH-L1 concentration was significantly higher for tumour sizes above 30 mm than for smaller-sized tumours. In addition, 20S proteasome and UCH-L1 were rated for their predictive value. Proteasome appears to correlate with aggressiveness (sensitivity 82%, specificity 42%) and UCH-L1 with invasiveness (75.77%).

Interestingly, the studies showed that the proteasome/UCH-L1 ratio in serum may be a useful additional marker. The cut-off value for ROC was calculated at 0.09.

It may be speculated that UCH-L1 is a stronger bladder cancer predictor: a higher concentration of this marker relative to the 20S proteasome increases the probability of cancer. This is in line with previous considerations of the role of UCH-L1 in the oncogenic process.

The results of this study may be useful in better understanding the biology of TCC tumours and in improving therapeutic methods. The 20S proteasome has been extensively explored as a drug target. Selective inhibition of the 20S proteasome has demonstrated a therapeutic benefit. The 20S proteasome inhibitor bortezomib has been used in lymphoma [52] and myeloma [53] treatment. This confirms the importance of investigations of the UPS in relation to cancer.

The SPRi biosensors discussed here allow the detection of 20S proteasome and UCH-L1 in serum and urine. They are a simple and cost-effective tool for determining the concentration of these biomolecules. The use of SPRi biosensors makes it possible to analyse complex samples without special procedures for their preparation. The biomolecules can be quantified without the need for labelling. SPRi may therefore be an attractive alternative to conventional testing.

The investigated UPS and DUB components do not appear to be highly specific markers of TCC but may be helpful in improving assessments of the progress of the disease and in making optimal therapeutic decisions.

**Author Contributions:** Conceptualisation, A.S., T.G. and E.G.; methodology, A.S. and E.G.; data collection, A.S. and T.G.; statistical analysis, A.S. and T.G.; data interpretation, A.S., E.G. and T.G.; writing—original draft preparation, A.S. and T.G.; writing—review and editing, E.G., A.S. and T.G.; visualisation, A.S. and T.G.; supervision, E.G. All authors have read and agreed to the published version of the manuscript.

**Funding:** This article has received financial support from the Polish Ministry of Science and Higher Education under a subsidy for maintaining the research potential of the Faculty of Chemistry, University of Bialystok (Grant No. BST 162/2018).

**Institutional Review Board Statement:** The study was conducted according to the guidelines of the Declaration of Helsinki and approved by the Bioethics Committee at the Medical University of Bialystok, nr R-I-002/409/2014.

**Informed Consent Statement:** Informed consent was obtained from all subjects involved in the study.

**Data Availability Statement:** Data available on request due to restrictions privacy.

**Conflicts of Interest:** The authors declare that they have no known competing financial interests or personal relationships that could have appeared to influence the work reported in this paper.

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
