# Peer review of "Application of SPRi Biosensors for Determination of 20S Proteasome and UCH-L1 Levels in the Serum and Urine of Transitional Bladder Cancer Patients"

_applsci, doi:10.3390/app11177835_

Round 1

Reviewer 1 Report

Report on Appl. Sci. 1318240, by Anna Sankiewicz, et al.:

            The authors propose an application of Surface Plasmon Resonance Imaging (SPRI) based biosensors for the detection of 20S proteasome and Ubiquitin Carboxy-Terminal Hydrolase L1 (UCH -L1) in the blood serum and urine of patients with Transitional Cell Carcinoma (TCC).  In this work, the author claims that the proposed scheme can be of the tools used in the investigation of cancer biomarkers.

           I feel that this work addresses an important topic and represents an advance over previous work.  I recommend publication after a few minor points are considered.

(1)   First of all, this technique is based on the SPRI. So, it would help the readers to give a very brief summary of the technique and also how the signals were measured.

(2)   The author says that the SPRI signal is measured at a constant angle of incident light.  It will help the readers if they explain why that constant angle was chosen and the effect of changing the angle.

(3) Also, it will be great if the authors can explain how their results compare with the conventional techniques.

Author Response

(1)   First of all, this technique is based on the SPRI. So, it would help the readers to give a very brief summary of the technique and also how the signals were measured.

(2)   The author says that the SPRI signal is measured at a constant angle of incident light.  It will help the readers if they explain why that constant angle was chosen and the effect of changing the angle.

The following text has been added to the introduction – page 5:

The SPR (Surface Plasmon Resonance) method is an optical technique that measures the changes of refractive index caused by molecules bound to a metal surface. Conventional SPR sensors measure the reflectance as a function of the angle of incident light.

The SPR Imaging version (SPRi) eliminates the complexity of scanning of the angle. The measurements are made at a particular angle of incident light. The reflected light is collected using a CCD and presented as an image. The angle at which the measurement is carried out lies in the region of linear decrease in reflectance. The changes in light intensity are proportional to the mass of biomolecules attached to the biosensor surface.

 A very important part of an SPRi system is a sensor chip with a sensitive recognition element. This is usually a glass chip coated with an inert metal (e.g. gold with thickness 50 nm).

The immobilization of biocomponents on a chemically modified gold layer takes place via covalent bonds, hydrophobic interaction, or adsorption. This is an important step in the manufacture of a biosensor, because it has an influence on the biosensor’s efficiency.

(3) Also, it will be great if the authors can explain how their results compare with the conventional techniques.

We do not have data available on measurements corresponding to those presented in the paper and obtained by methods other than the use of SPR biosensors. The biosensors used, presented in earlier publications [29, 30], were validated through comparison with literature data or with data obtained from ELISA tests.

Reviewer 2 Report

In this paper, the authors used Surface Plasmon Resonance imaging (SPRi) to determine 20S proteasome and UCH-L1 concentrations in the blood serum and in the urine of patients with bladder cancer. The work is interesting, but however I do have some remarks. Under these circumstances, I would recommend the publication of this manuscript in “Applied Sciences” only after a minor revision of the content. My comments are appended below.

1. In general the English throughout the entire manuscript needs proof-reading.

Some examples: “In this work we propose an application Surface Plasmon Resonance Imaging (SPRI)-based biosensors for the detection of […]”

“[…] can be of tools used in the investigation of cancer biomarkers.”

“SPRi biosensors represent the one of type of optical biosensor.”

2. Please use the same abbreviation for Surface Plasmon Resonance imaging (SPRi) through all manuscript

3. In the abstract is mentioning suddenly “These two biosensors are suitable for determination the 20S proteasome and UCH-L1”, what are the two biosensors? Please be more specific, otherwise is confusing.

4. The authors mention that the most common method for detection of 20S proteasome and UCH-L1 is ELISA, and that SPRi can be also used. Within this paragraph is missing a short comparison between the 2 techniques, what are the advantages of SPRi compared to ELISA?

5. Regarding the introduction part, the novelty of this research work is not sufficiently highlighted. The authors are detecting 20S proteasome and UCH-L1 using SPRi for the first time or there are in literature some other studies? Please briefly compare your results with literature, to underline the novelty of your work.

6. In Figure 2 there are data shown only for proteasome 20S and UCH-L1 concentrations in serum, not urine. I think Figures 2 B,D are for urine, no?

7. The authors mentioned that the serum 20S proteasome concentration and the UCH-L1 concentration are elevated also in the case of other types of cancer. Therefore, if a patient has this concentrations elevated he won’t know what type of cancer he has, just that he has cancer?

Author Response

  1. In general the English throughout the entire manuscript needs proof-reading.

Some examples: “In this work we propose an application Surface Plasmon Resonance Imaging (SPRI)-based biosensors for the detection of […]”

“[…] can be of tools used in the investigation of cancer biomarkers.”

“SPRi biosensors represent the one of type of optical biosensor.”

The manuscript has been checked again by a native speaker.

  1. Please use the same abbreviation for Surface Plasmon Resonance imaging (SPRi) through all manuscript

The abbreviation for Surface Plasmon Resonance imaging (SPRi) has been corrected throughout the manuscript.

  1. In the abstract is mentioning suddenly “These two biosensors are suitable for determination the 20S proteasome and UCH-L1”, what are the two biosensors? Please be more specific, otherwise is confusing.

The following text has been inserted into the abstract:

The SPRi biosensor sensitive to 20S proteasome using PSI inhibitor as the receptor, and the SPRi biosensor sensitive to the UCH-L1 protein using the protein-specific antibody as the receptor are suitable for the determination of 20S proteasome and UCH-L1 in body fluids, and can serve as useful tools in the investigation of cancer biomarkers.

  1. The authors mention that the most common method for detection of 20S proteasome and UCH-L1 is ELISA, and that SPRi can be also used. Within this paragraph is missing a short comparison between the 2 techniques, what are the advantages of SPRi compared to ELISA?

The following text has been added to the introduction – page 6:

Because the levels of 20S proteasome and UCH-L1 can be detected in serum [7, 10, 27, 28], they can be used as potential cancer biomarkers. ELISA is the preferred method for the determination of 20S proteasome and UCH-L1 concentrations. Although it is a reliable technique with good sensitivity and selectivity, it is relatively time- and labor-consuming. SPRi may be an alternative technique to traditional immunobased assays like ELISA. Both methods have advantages, but also disadvantages. SPR has been found to be comparable to ELISA in terms of sensitivity and specificity. In SPR, 20S proteasome and UCH-L1 concentrations can be detected in real time using only a single antibody or inhibitor [27, 29-31]. Unlike ELISA, SPRi does not require a label or an additional reagent. An SPR biosensor requires very low amounts of reagents and samples, and does not require any special treatment [32-35]. It enables the rapid and precise determination of 20S proteasome and UCH-L1 concentrations.

  1. 5. Regarding the introduction part, the novelty of this research work is not sufficiently highlighted. The authors are detecting 20S proteasome and UCH-L1 using SPRi for the first time or there are in literature some other studies? Please briefly compare your results with literature, to underline the novelty of your work.

The following text has been added to the introduction – page 5:

To our knowledge, this is the first paper presenting 20S proteasome and UCH-L1 protein data obtained from measurements in the serum and urine of bladder cancer patients.

The following text has been added to the discussion section – page 20-21:

The measurements presented in the paper are the first available data on concentrations of 20S proteasome and UCH-L1 protein, and their mutual correlation, in the serum and urine of bladder cancer patients.

  1. In Figure 2 there are data shown only for proteasome 20S and UCH-L1 concentrations in serum, not urine. I think Figures 2 B,D are for urine, no?

Figure 2 has been corrected

  1. The authors mentioned that the serum 20S proteasome concentration and the UCH-L1 concentration are elevated also in the case of other types of cancer. Therefore, if a patient has this concentrations elevated he won’t know what type of cancer he has, just that he has cancer?

So far, unfortunately, there is no known fully specific marker for a particular type of cancer. Therefore, laboratory diagnostics must be synchronized with imaging diagnostics and the patient’s clinical data. Broad possibilities in this regard are provided by research into so-called diagnostic panels, i.e. a specific number of markers and their mutual correlations. The algorithms used, for example in the ROMA test, enable more precise determination of the type of neoplasm or other non-neoplastic disease entity.